# Synthesis of Nanostructure In$_x$Ga$_{1-x}$N Bulk Alloys and Thin Films for LED Devices

**Abd El-Hady B. Kashyout [1], Marwa Fathy [1], Sara Gad [1,*], Yehia Badr [2] and Ahmed A. Bishara [3]**

[1] Electronic Materials Research Department, Advanced Technology and New Materials Research Institute, City of Scientific Research and Technological Applications (SRTA-City), Alexandria 21934, Egypt; akashyout@srtacity.sci.eg (A.E.-H.B.K.); mbahnase@srtacity.sci.eg (M.F.)

[2] National Institute of Laser Enhanced Science, Laser Interaction with Matter Department, Cairo University, Cairo 94142, Egypt; ybadr@niles.edu.eg

[3] Department of physics, Faculty of science, Alexandria University, Alexandria 21543, Egypt; Ahmed.bishara@alex.edu.eg

* Correspondence: sarapgad@gmail.com or egad@srtacity.sci.eg

**Abstract:** In this study, we investigated an innovative method for the fabrication of nanostructure bulk alloys and thin films of indium gallium nitride (In$_x$Ga$_{1-x}$N) as active, thin films for light-emitting diode (LED) devices using both crystal growth and thermal vacuum evaporation techniques, respectively. These methods resulted in some tangible improvements upon the usual techniques of In$_x$Ga$_{1-x}$N systems. A cheap glass substrate was used for the fabrication of the LED devices instead of sapphire. Indium (In) and Gallium (Ga) metals, and ammonia (NH$_3$) were the precursors for the alloy formation. The alloys were prepared at different growth temperatures with compositions ranging from $0.1 \leq x \leq 0.9$. In$_x$Ga$_{1-x}$N alloys at $0.1 \leq x \leq 0.9$ had different crystallinities with respect to X-Ray diffraction (XRD) patterns where the energy bandgap that was measured by photoluminescence (PL) fell in the range between 1.3 and 2.5 eV. The bulk alloys were utilized to deposit the thin films onto the glass substrate using thermal vacuum evaporation (TVE). The XRD thin films that were prepared by TVE showed high crystallinity of cubic and hexagonal structures with high homogeneity. Using TVE, the In$_x$Ga$_{1-x}$N phase separation of $0.1 \leq x \leq 0.9$ was eliminated and highly detected by XRD and FESEM. Also, the Raman spectroscopy confirmed the structure that was detected by XRD. The FESEM showed a variance in the grain size of both alloys and thin films. The In$_x$Ga$_{1-x}$N LED device with the structure of glass/GaN/n-In$_{0.1}$Ga$_{0.9}$N:n/In$_{0.1}$Ga$_{0.9}$N/p-In$_{0.1}$Ga$_{0.9}$N:Mg was checked by the light emitted by electroluminescence (EL). White light generation is a promising new direction for the fabrication of such devices based on In$_x$Ga$_{1-x}$N LED devices with simple and low-cost techniques.

**Keywords:** LED; In$_x$Ga$_{1-x}$N; nanostructures

## 1. Introduction

In$_x$Ga$_{1-x}$N is an exciting material because it has a bandgap that spans from the ultraviolet to the visible spectrum, so the material can be tuned and changed according to its composition. According to Vegard's law, it is possible to change the energy value of the band gap and it can range from 3.4 eV (GaN) to 0.7 eV (InN) [1] by varying the indium composition at $0.1 \leq x \leq 0.9$ into the In$_x$Ga$_{1-x}$N compound. The previous research for bulk alloys depended on high-cost equipment techniques such as high nitrogen pressure solution growth, ammonothermal growth, Na flux growth and crystal growth [2–4]. Until the year 1969, there was no significant advancement in the growth of these compounds, until the growth of a GaN epilayer that was reported by hydride vapor phase epitaxial (HVPE), which is a suitable growth method for producing bulk materials [5]. Metal organic chemical

vapor deposition (MOCVD) was first developed for the growth of GaN [6]. Both HVPE [7,8] and MOCVD [9,10] techniques were used to grow GaN directly on sapphire. Also, the plasma-assisted molecular beam epitaxy (PAMBE) was used to fabricate $In_xGa_{1-x}N$/GaN hetero-structure nano-rods without embedded quantum-well QW-active layers [11]. The difficulties in $In_xGa_{1-x}N$ growth are mainly due to very high equilibrium vapor pressures (EVPs) of nitrogen over InN and a large lattice mismatch between InN and GaN. Additionally, the large lattice mismatch between InN and GaN resulted in highly strained $In_xGa_{1-x}N$ alloys, which means that phase separation is a major concern.

In fact, the phase separation could be driven on the surface of $In_xGa_{1-x}N$ layers, especially during the growth of MOCVD or molecular beam epitaxy (MBE). The majority of the III-V ternary and quaternary alloys is predicted to be thermodynamically unstable and shows a tendency towards clustering and phase separation [12,13]. Ho et al. studied the temperature dependence of the binodal and spinodal lines in the $In_xGa_{1-x}N$ system using a modified valence-force-field model. Phase separation was observed by Osamura et al. in 1975 by annealing polycrystalline $In_xGa_{1-x}N$ samples in an argon atmosphere between 600 and 700 °C [14]. More recently, Singh et al. reported experimental evidence of phase separation in molecular beam epitaxy (MBE)-grown $In_xGa_{1-x}N$ using X-ray diffraction and optical absorption measurement [15]. This phenomenon affects the structure of the materials and consequently their performance.

In this study, we observed the phase separation of $In_{0.3}Ga_{0.7}N$ alloys while they had disappeared in other compositions and all the thin films. So, we state a simple crystal growth technique and thermal vacuum evaporation (TVE) for the fabrication of $In_xGa_{1-x}N$ (x = 0.1, 0.3, 0.5, 0.7 and 0.9) bulk alloys and their relevant thin films, respectively. The crystal growth technique uses ammonia gas with Ga and In metals in a closed evacuated silica tube and the produced alloys were utilized for thin film evaporation via TVE. $In_xGa_{1-x}N$ LED devices were structured from an n-type layer of $In_{0.1}Ga_{0.9}N$ doped with Zn metal and a p-type layer of $In_{0.1}Ga_{0.9}N$ doped with Mg metal. The morphology of the $In_xGa_{1-x}N$ surfaces was examined by implementing field emission scanning electron microscopy (FESEM). The crystalline phase of $In_xGa_{1-x}N$ bulk alloys and thin films was analyzed by X-ray diffraction (XRD). The optical properties were examined by photoluminescence (PL). The compositions of the bulk alloys and thin films were determined using energy dispersive X-ray spectroscopy technique (EDX) which was attached to the FESEM unit. By using scanning electroluminescence (EL) spectroscopy, we investigated the emitted light from the $In_xGa_{1-x}N$ LED devices.

## 2. Materials and Methods

### 2.1. Materials

Gallium metal (99.999%), Indium metal (99.999%), Zinc wire (99.999%), Gold target for metallization, and Magnesium granules (99.999%) were purchased from Sigma Aldrich, Chemical Co.; USA. Ammonia gas 99.999% was purchased from (Alexandria for gases). The glass substrates were washed and sonicated in distilled water for 30 min.

### 2.2. Bulk Alloy Preparation by the Crystal Growth Technique

Appropriate weights of In and Ga metals were mixed and put in a clean silica ampoule closed from one side in order to produce $In_xGa_{1-x}N$ bulk alloys with different compositions (x = 0.1, 0.3, 0.5, 0.7 and 0.9). The ampoule was then evacuated from the other side at a pressure of $10^{-4}$ Torr and sealed by welding the narrow nick which was made to insert $NH_3$, as shown in Figure 1a. The flow rate of ammonia was 0.1 sccm for 7 min. The growth of the bulk alloys was prepared with various compositions of 0.1, 0.3, 0.5, 0.7, and 0.9 for In metal, at various temperatures of 750 °C, 950 °C and 1150 °C. The resulting alloy composite was cooled slowly to ambient temperature in order to prevent segregation, as shown in Figure 1b. The reaction schedule can be schematically represented in Figure 2 and explained by these equations as follows:

$$Ga \text{ (Jelly)} + NH_3 \longrightarrow GaN(s) + 3/2H_2 \tag{1}$$

$$In \text{ (Metal)} + NH_3 \longrightarrow InN(s) + 3/2H_2 \tag{2}$$

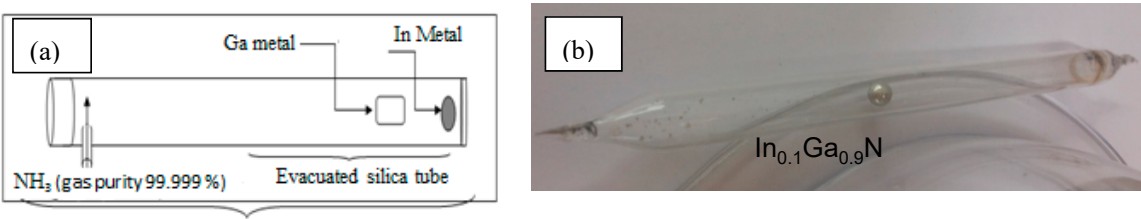

**Figure 1.** (**a**) Experimental setup. The tube contains In and Ga metals and $NH_3$ gas. (**b**) The resulting indium gallium nitride ($In_xGa_{1-x}N$) alloy after reaction.

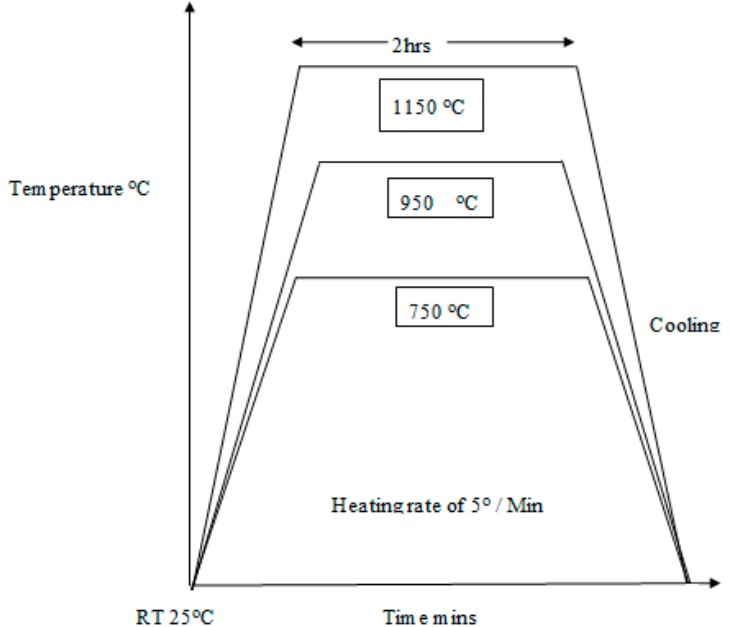

**Figure 2.** Schematic diagram for the growth process of the $In_xGa_{1-x}N$ alloy under different temperatures.

*2.3. Preparation of Doped Alloys by the Crystal Growth Technique*

2.3.1. Preparation of $In_{0.1}Ga_{0.9}N$: Mg [p-type]

The weighted elements, i.e., In and Ga metals, were put in a silica tube under $NH_3$ gas with the addition of 0.02% Mg as a p-type dopant to prepare $In_{0.1}Ga_{0.9}N$: Mg under a growth temperature of 750 °C with a temperature rise of 5 °C/min. After reaching 750 °C, it was raised to 800 °C and stayed at this temperature for 30 min, then it was cooled to 530 °C for 20 min with the muffle off, allowing the sample to be cooled.

2.3.2. Preparation of $In_{0.1}Ga_{0.9}N$: Zn [n-type]

The weighted elements, i.e., In and Ga metals, were put in a silica tube under $NH_3$ gas with 0.02% Zn. The elements reached 800 °C and stayed at this temperature for 1 h and then the muffle furnace was cooled to room temperature.

### 2.3.3. Preparation of Bulk Buffer GaN Alloy

The weighed gallium element with 0.1 gm. was put in the vacuumed silica tube filled with $NH_3$ gas. The tube was kept at 950 °C for 2 h then the muffle was cooled to room temperature.

## 3. Results

### *3.1. X-ray Diffraction*

#### 3.1.1. For $In_xGa_{1-x}N$ Alloys Prepared at Different Compositions

Figure 3 shows XRD patterns for $In_xGa_{1-x}N$ alloys prepared at different compositions of In (x = 0.1, 0.3 0.5, 0.7 0.9) at a heating temperature of 950 °C for 2 h. As the indium composition increases, a shift in the 2θ value from 33° to 23° and broadening of the main peak is observed. This confirms a change of indium content according to the corresponding 2θ value of the same crystallographic planes (002), (100) and (101) with respect to that of InN (JCPDS Card No: 01-088-2365) and GaN (JCPDS No: 00-052-0871) [16]. Therefore, the formation of the $In_xGa_{1-x}N$ phase is obtained in the range of 2θ between approximately 31° and 37° [17]. When the In composition increases from 0.1 to 0.5, the same diffraction peaks are observed between 30.73° and 37.77°, which matches with some previous studies [18,19]. This indicates a good structure quality and orientations of $In_xGa_{1-x}N$ alloys. As the indium composition increases to 0.7 and 0.9, the peaks shift to InN with a wurtzite structure (hexagonal). In $In_{0.3}Ga_{0.7}N$, there is an additional peak at 67.3° which belongs to the GaN structure with respect to that of InN (JCPDS Card No: 00-050-0792). It refers to the phase separation that is only found in this composition [20]. The broadening of the peaks might arise from several factors such as the distribution of bond lengths for a random alloy size-dependent broadening, the large size of the X-ray beam spot across the composition gradient and the inhomogeneity of the ternary $In_xGa_{1-x}N$ alloys [21].

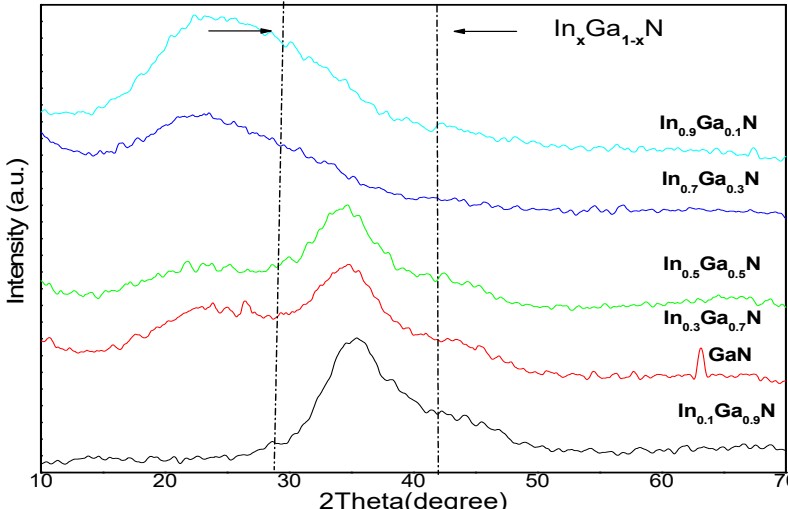

**Figure 3.** X-Ray diffraction (XRD) patterns of the bulk $In_xGa_{1-x}N$ alloy at different In compositions (x = 0.1, 0.3, 0.5, 0.7, 0.9) prepared at 950 °C for 2 h.

#### 3.1.2. $In_xGa_{1-x}N$ Thin Film Deposited by TVE at Different Compositions

Figure 4 shows the XRD patterns of $In_xGa_{1-x}N$ thin films deposited by thermal vacuum evaporation (TVE). $In_xGa_{1-x}N$ thin films grown on glass at x = 0.1, 0.3, 0.9 have diffraction peaks of 33°, 32° and 33°, respectively, with a hexagonal structure as matched with ref [22]. At x = 0.5 and 0.7, the diffraction angle is obtained at 2θ of 38° with the cubic structure that matched with ref [23]. No extra peaks were observed in all samples and it may mean that no In droplets or phase separation was observed from the $In_xGa_{1-x}N$ layers [24]. This indicates a good structure quality and orientation.

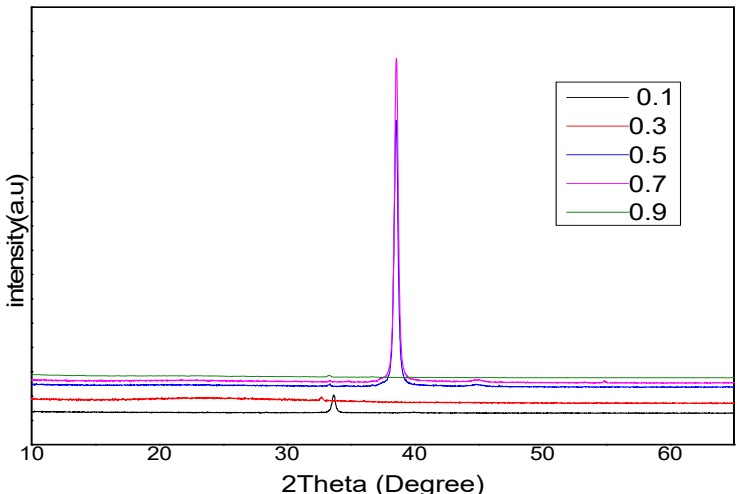

**Figure 4.** XRD patterns of $In_xGa_{1-x}N$ (x = 0.1, 0.3, 0.5, 0.7 and 0.9) thin films deposited on glass substrates using thermal vacuum evaporation (TVE).

### 3.2. Compositional Analysis

#### 3.2.1. Bulk Alloys and Thin Film at Different Compositions Prepared by TVE

The comparison between the theoretical calculated composition of $In_xGa_{1-x}N$ bulk alloys, and thin films weighed before preparation, and the measured composition by the EDX is represented in Table 1. It is noted that there is no great difference of values between thermal vacuum deposited thin films and bulk alloys in their chemical composition with respect to the nominal weighted materials. This indicates that the composition is well controlled by the used preparation techniques. The ratio of nitrogen in both bulk and thin films shows that there is no change across the two techniques except for bulk composition at x = 0.5 and thin film at x = 0.1.

**Table 1.** Energy dispersive X-ray spectroscopy technique (EDX) measurements of $In_xGa_{1-x}N$ bulk alloys and thin films.

| $In_xGa_{1-x}N$ | Bulk Alloys | | | Thin Films | | |
|---|---|---|---|---|---|---|
| | In% | Ga% | N% | In% | Ga% | N% |
| $In_{0.1}Ga_{0.9}N$ | 11.46 | 86.02 | 2.52 | 17.41 | 60.73 | 21.85 |
| $In_{0.3}Ga_{0.7}N$ | 27.3 | 64.2 | 8.5 | 31.06 | 63.78 | 5.17 |
| $In_{0.5}Ga_{0.5}N$ | 42.90 | 40.70 | 16.4 | 48.05 | 44.77 | 7.18 |
| $In_{0.7}Ga_{0.3}N$ | 67.78 | 25.51 | 6.7 | 60.67 | 32.65 | 6.68 |
| $In_{0.9}Ga_{0.1}N$ | 88.02 | 9.46 | 2.52 | 82.67 | 13.41 | 3.92 |

### 3.3. Morphology

#### 3.3.1. Bulk Alloys with Different Compositions Prepared by Crystal Growth

Figure 5 shows FESEM images for bulk alloys at $In_xGa_{1-x}N$ heated at 950 °C (0.1≤ x ≤ 0.9) and it demonstrates clearly distinct nanowires in the bulk alloys with increasing indium incorporation.

The nanowires show a high degree of order. As the indium composition increases, the widening of the nanowire's diameter and length decreases significantly, as seen in Table 2.

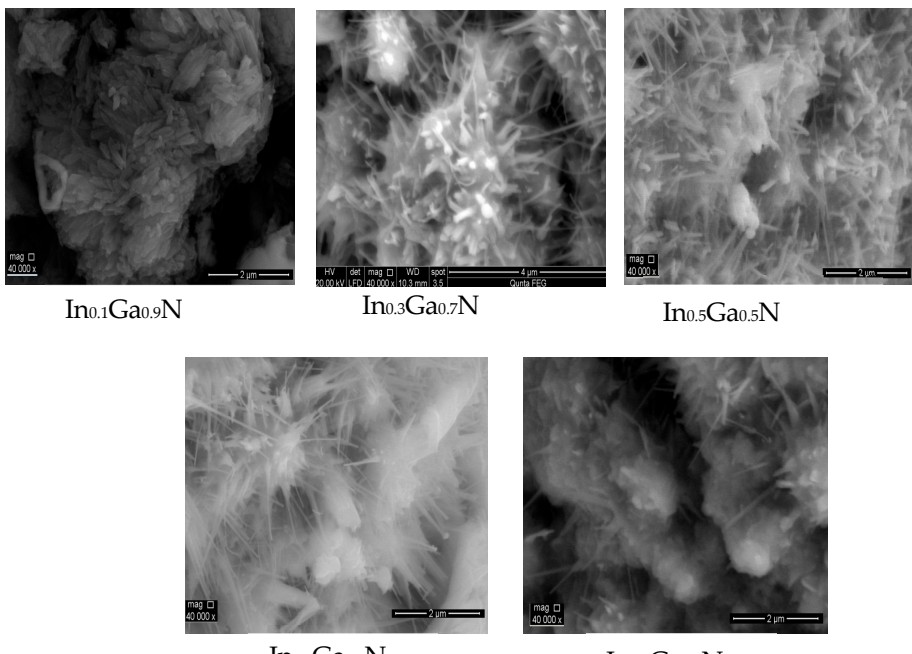

In$_{0.1}$Ga$_{0.9}$N      In$_{0.3}$Ga$_{0.7}$N      In$_{0.5}$Ga$_{0.5}$N

In$_{0.7}$Ga$_{0.3}$N      In$_{0.9}$Ga$_{0.1}$N

**Figure 5.** Field emission scanning electron microscopy (FESEM) for bulk alloys at In$_x$Ga$_{1-x}$N heated at 950 °C ($0.1 \leq x \leq 0.9$).

**Table 2.** Measurements of nanowire diameter and length from FESEM for In$_x$Ga$_{1-x}$N bulk alloys.

| Composition of In$_x$Ga$_{1-x}$N Bulk Alloy (x%) | Bulk Alloy Deposited on Glass | |
| :---: | :---: | :---: |
| | Length (nm) | Diameter (nm) |
| 0.1 | 341 | 69 |
| 0.3 | 176 | 49 |
| 0.5 | 153 | 9.1 |
| 0.7 | 142 | 7.7 |
| 0.9 | 104 | 4.25 |

### 3.3.2. Thin Films Prepared by TVE at Different Compositions Deposited on Glass

Figure 6 shows the FESEM for thin films of In$_x$Ga$_{1-x}$N deposited by thermal vacuum evaporation. It is noted that surface morphologies of the In$_x$Ga$_{1-x}$N films with different indium compositions have a smooth surface with homogenous nano-spherical sized particles of the films. At x = 0.1, the thin film is denser than the other compositions and exhibits good surface morphologies. The average diameter of the particle size decreases as the In composition increases.

Table 3 shows the particle size that was measured from FESEM images which has little difference compared to the size calculated from Scherer's equation. It ranged from about 35 nm at x = 0.1 to 22 nm at x = 0.9. The observed larger grain sizes are composed of fine nanoparticles and some grains of about 500 nm at x = 0.1, 0.3, which decreased to about 290 nm at x = 0.5, 0.7 and 0.9.

**Table 3.** The particle size measurements from FESEM and Scherer's equation for thin films deposited on a glass substrate.

| Composition of In$_x$Ga$_{1-x}$N Thin Films | Particle Size (nm) from FESEM | Crystallite Size (nm) from XRD |
| :---: | :---: | :---: |
| 0.1 | 35.1 | 32.23 |
| 0.3 | 28.8 | 25.04 |
| 0.5 | 27.1 | 24.73 |
| 0.7 | 22.9 | 24.39 |
| 0.9 | 22.1 | 20.55 |

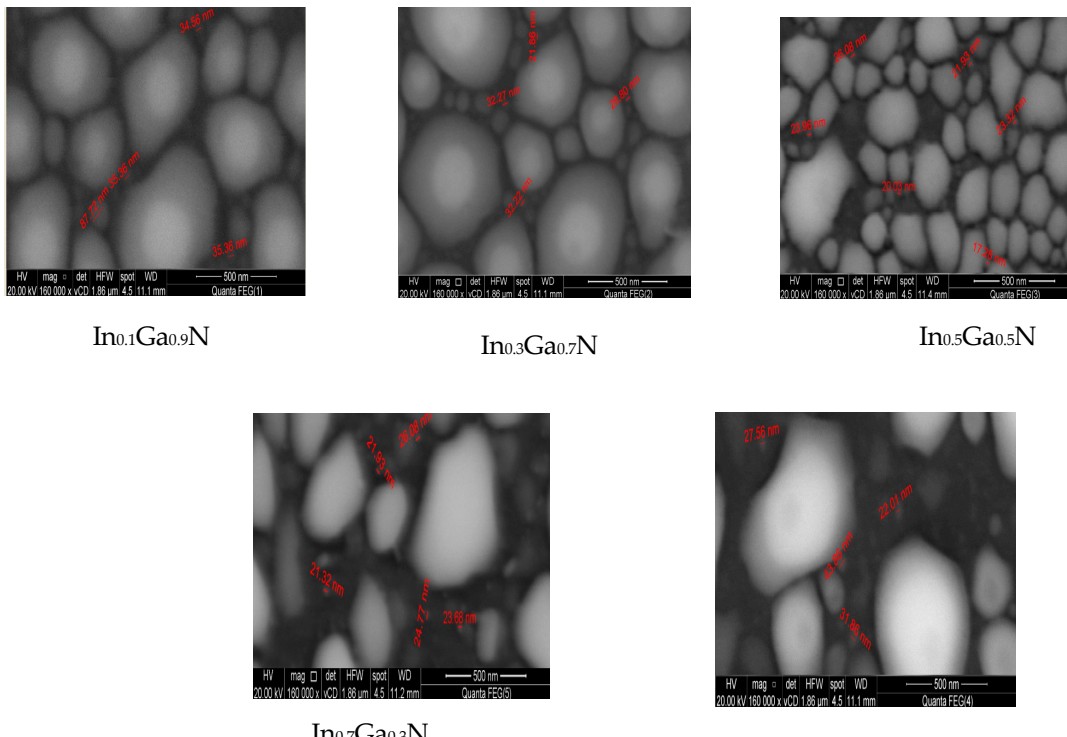

In<sub></sub>$In_{0.1}Ga_{0.9}N$　　　　　　　$In_{0.3}Ga_{0.7}N$　　　　　　　$In_{0.5}Ga_{0.5}N$

$In_{0.7}Ga_{0.3}N$　　　　　　　$In_{0.9}Ga_{0.1}N$

**Figure 6.** FESEM for $In_xGa_{1-x}N$ thin films prepared with TVE for the alloy prepared at 950 °C.

### 3.4. Optical Properties for Thin Films

Photoluminescence (PL) Thin Films Prepared by Thermal Vacuum Evaporation

The PL emission for thin films of $In_xGa_{1-x}N$ deposited by thermal vacuum evaporation is shown in Figure 7. No extra PL emission peaks are observed with different composition contents. The observed optical emission at room temperature is attributed to radiative band-to-band transitions. A red shift of the optical emission with increasing Indium content is observed [25] and this ranged from 2.6 nm to 2.4 nm. Also, the widening of the peaks for compositions of x = 0.3, 0.7 may be due to the lowest states of the conduction band being blocked, a phenomenon known as the Burstein–Moss effect [25], which occurs with an increase in the carrier density. The narrowing of the peaks for compositions 0.1, 0.5, and 0.9 may be due to the crystallinity and homogeneity of the thin films, as shown by XRD data [25]. In order to determine the band gap of all samples, each PL spectrum was fitted by a Gaussian distribution function, as shown in Figure 7. The band gap values were directly obtained from the maximum of the PL peaks, as shown in Table 4. The band gap values obtained from the PL measurements were of the same order of magnitude as is found in a variety of $In_xGa_{1-x}N$ alloys [26–29]. In the case of bulk alloys, the bandgap values at an In content of 0.1 and 0.3 were compatible with the $In_xGa_{1-x}N$ system. The thin films prepared by TVE and the small bandgap values at x = 0.5, 0.7 and 0.9, as shown in Table 4, refer to a possible phase separation in the bulk alloys which is previously explained by the XRD data presented in Figure 3.

The calculation of the bowing parameter, as shown in Table [3.1], depends on Vegard's law [30]:

$$Eg\ (In_xGa_{1-x}N) = Eg\ GaN\ (1-x) + Eg\ InN(x) - bx\ (1-x) \tag{3}$$

where Eg(GaN) = 3.4 eV, Eg(InN) = 0.7 eV and b = bowing parameter

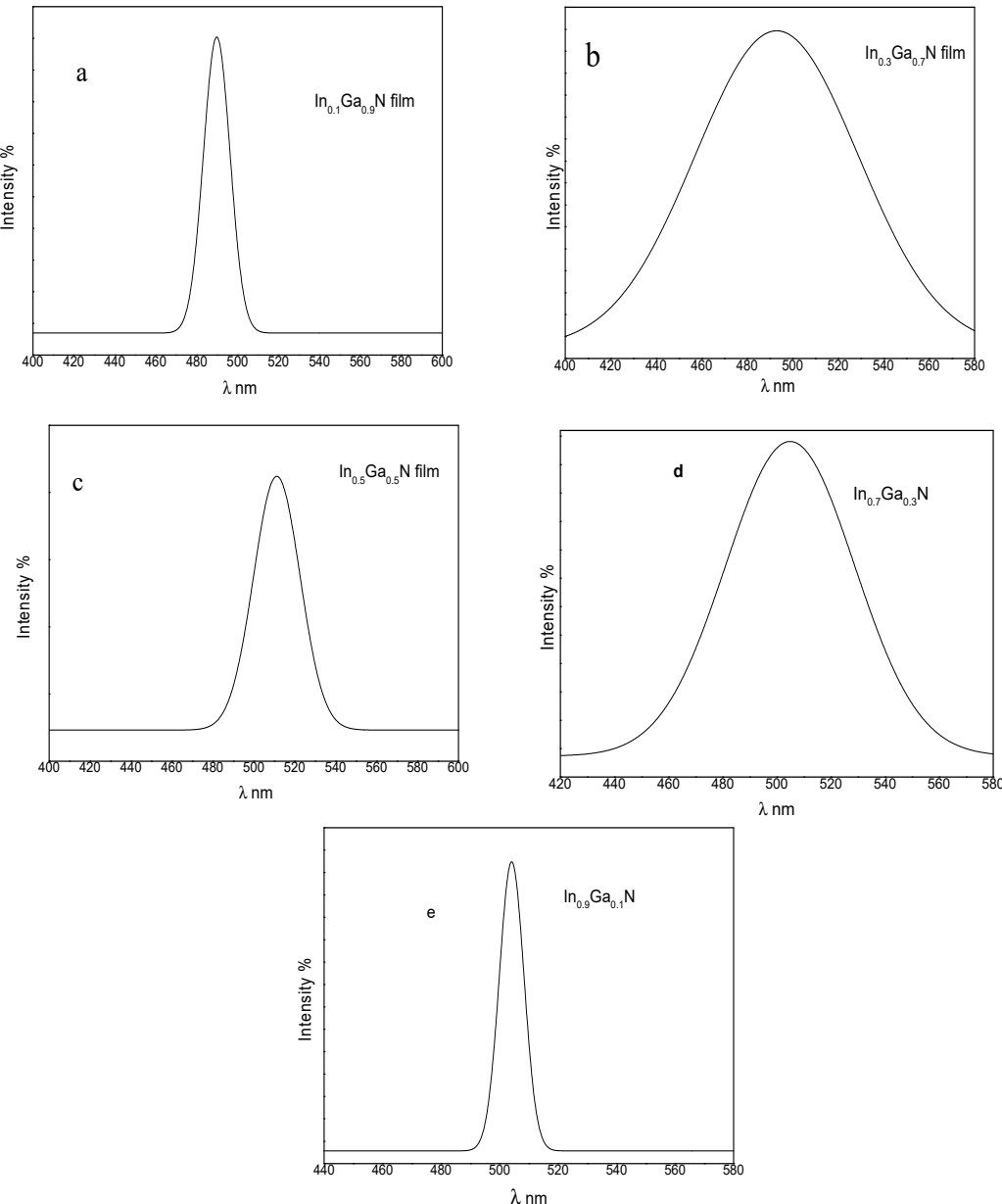

**Figure 7.** PL-spectra of $In_xGa_{1-x}N$ thin films deposited at $(0.1 \leq x \leq 0.9)$.

**Table 4.** Optical properties of $In_xGa_{1-x}N$ as bulk alloys and thin films.

| Composition (In) | Bulk Alloys | | Thin Films | |
|---|---|---|---|---|
| | Bowing Parameter (eV) | Bandgap (eV) $In_xGa_{1-x}N$ | Bowing Parameter | Bandgap(eV) $In_xGa_{1-x}N$ |
| 0.1 | 2.89 | 2.65 | 1.24 | 2.61 |
| 0.3 | 2.12 | 2.16 | 1.04 | 2.51 |
| 0.5 | 1.87 | 1.78 | 0.97 | 2.45 |
| 0.7 | 1.67 | 1.67 | 0.85 | 2.44 |
| 0.9 | 1.33 | 1.49 | 0.77 | 2.42 |

In bulk alloys, the bowing calculations show large values at (x = 0.1 and 0.3), except for (x = 0.5, 0.7 and 0.9) which have small values. As shown in Table 4, the highest bandgap bowing parameter is (b = 2.89 eV) for bulk alloys. In thin films prepared by TVE, the bowing values are decreased as the composition increases. Since the $In_xGa_{1-x}N$ alloys are highly disordered crystal systems, there are large

differences in the lattice constants between GaN and InN [31]. Due to this disordered crystal geometry, it is possible to undergo metamorphosis of physical properties, such as band-structure deformation.

### 3.5. Raman Spectroscopy

#### 3.5.1. $In_{0.1}Ga_{0.9}N$ Thin Films Deposited by Thermal Vacuum Evaporation

It has been established that the $In_xGa_{1-x}N$ thin films exhibit two modes of behavior, i.e., the Raman spectrum consists of the $A_1^{LO}$ and E2 lines which vary in frequency between those of the two binary compounds InN and GaN [32]. The Raman measurements of the $In_xGa_{1-x}N$ thin films were made at room temperature in the backscattering configuration. The spectrum recorded at the excitation wavelength is λexc = 530 nm. Raman spectra from $In_xGa_{1-x}N$ thin films (x = 0.1, 0.3, 0.5, 0.7 and 0.9) are displayed in Figure 8. The InN, GaN and $In_xGa_{1-x}N$ are stable in the hexagonal wurtzite phase, in which their scattering at the E2 and $A_1^{LO}$ phonons is allowed. The frequency values of GaN at $A_1^{LO}$ = 734 cm$^{-1}$ and E2 = 568 cm$^{-1}$, 570 cm$^{-1}$, while the frequency values of the InN $A_1^{LO}$ = 586 cm$^{-1}$, and E2 = 488cm$^{-1}$. Also, there is a phonon band in the frequency region of the $A_1^{LO}$ $In_xGa_{1-x}N$ vibrations (700–750 cm$^{-1}$). The observed behavior of the modes is reported in several previous works [33]. In (x = 0.3, 0.5, 0.7 and 0.9) compositions, there is a shift of the frequency and high broadening due to elastic strain between the thin film and glass substrate, and also due to high Indium content [34].

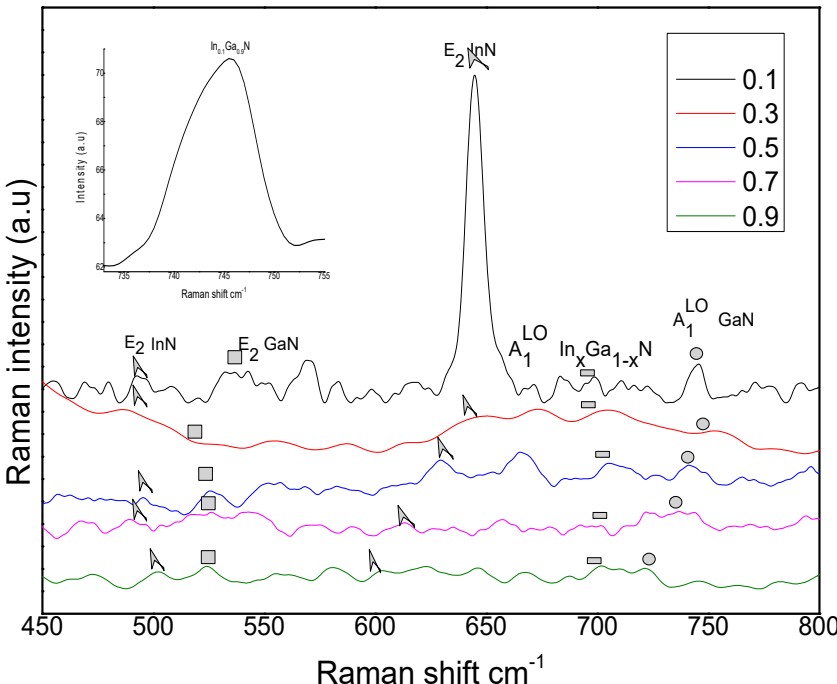

**Figure 8.** Raman spectroscopy of $In_xGa_{1-x}N$ thin film prepared by TVE at different x ratios.

### 3.6. Device Fabrication

#### 3.6.1. Device Characterizations

LED layered devices with a configuration of Glass/GaN/$In_{0.1}Ga_{0.9}N$:Zn/$In_{0.1}Ga_{0.9}N$/$In_{0.1}Ga_{0.9}N$: Mg were deposited on glass by thermal vacuum evaporation. The cross-sectional FESEM images of these devices are given in Figure 9. These images show the regular and uniform thickness of each layer, which is found to be 0.072 μm for the buffer layer (GaN), 1.07 μm for the n-type layer $In_{0.1}Ga_{0.9}N$, 1.18 μm for the active layer ($In_{0.1}Ga_{0.9}N$), and 0.167 μm for the p-type layer $In_{0.1}Ga_{0.9}N$. The FESEM results showed that the thicknesses of the GaN buffer layer were less than the values of all LED layers.

This structure generates a white light for the first time and is shown in Figure 10, which is taken by a digital camera during the light emission.

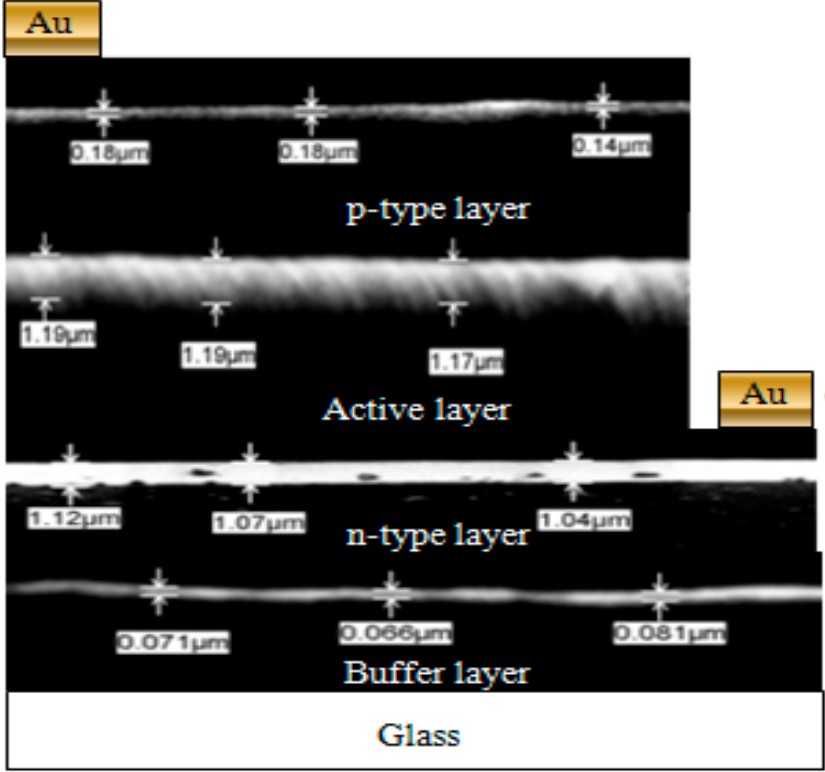

**Figure 9.** Cross-section of the $In_{0.1}Ga_{0.9}N$ LED.

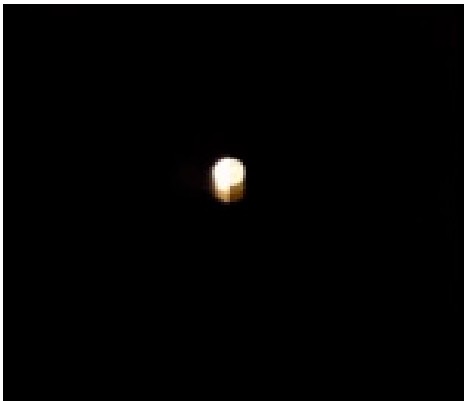

**Figure 10.** White light generated from the $In_{0.1}Ga_{0.9}N$ LED.

The EL spectra, as shown in Figure 11, show that $In_xGa_{1-x}N$ is shaped as a wide spectrum in the 300–900 nm range. The EL spectra for the LED structure contain glass/GaN/$In_{0.1}Ga_{0.9}N$: Zn/$In_{0.1}Ga_{0.9}N$/$In_{0.1}Ga_{0.9}N$: Mg coated by Au at both sides of p and n types, forming the $In_xGa_{1-x}N$ LED structure. The use of Gaussian line shapes allowed the discrimination of three overlapping luminescence-using bands with their maximums at 426 nm, 548 nm, and 599 nm respectively. The shorter wavelength in the range of 402 to 426 nm may be due to two reasons. Firstly, it may be due to the $In_{0.1}Ga_{0.9}N$ deposited on GaN which gives blue emission spectra. This blue emission of EL spectra suggests that the luminescence mechanism is in donor–acceptor transition in the $In_{0.1}Ga_{0.9}N$ active layer co-doped with Zn. The $In_{0.1}Ga_{0.9}N$ active layer of the blue spectra is doped with Zn. Thus, the Zn acceptor level is expected to exhibit Gaussian broadening and both the valence band and the conduction band will have band tails [35,36]. The shorter wavelength at 548 nm may be due to the p-type doping

which gives green emission [37]. The EL curve of the device shows a green peak emission, which implies that the recombination of holes and electrons occurs predominantly in the green light near the p-$In_{0.1}Ga_{0.9}N$ layer. This phenomenon indicates that the Mg doping concentration of the sample has little effect on hole injection into the LED structure. So, most holes are trapped in the green emission due to their lower mobility than that of electrons in the $In_{0.1}Ga_{0.9}N/GaN$. Due to Mg doping into the material, the recombination of holes and electrons occurred [38,39]. There is a yellow band at 560 nm [40] that may belong to GaN and a new red band near 620 nm which may be due to a recombination via deep levels in $In_{0.1}Ga_{0.9}N$. The violet band at 410 nm can be ascribed to a donor–acceptor pair recombination in the GaN. According to the $In_xGa_{1-x}N$ structure, a blue emission originating from a recombination in the $In_xGa_{1-x}N$: Mg deposited on the $In_xGa_{1-x}N$ layer and shifted toward a short wavelength (599 nm). Since the mobility of electrons is faster than holes, electrons are injected from the n- $In_xGa_{1-x}N$: Zn side, through the $In_xGa_{1-x}N$ layer, to the p- $In_xGa_{1-x}N$: Mg side and little recombination occurred in the n- $In_xGa_{1-x}N$ and $In_xGa_{1-x}N$ layers. Thus, the blue emission peak at 599 is observed [41].

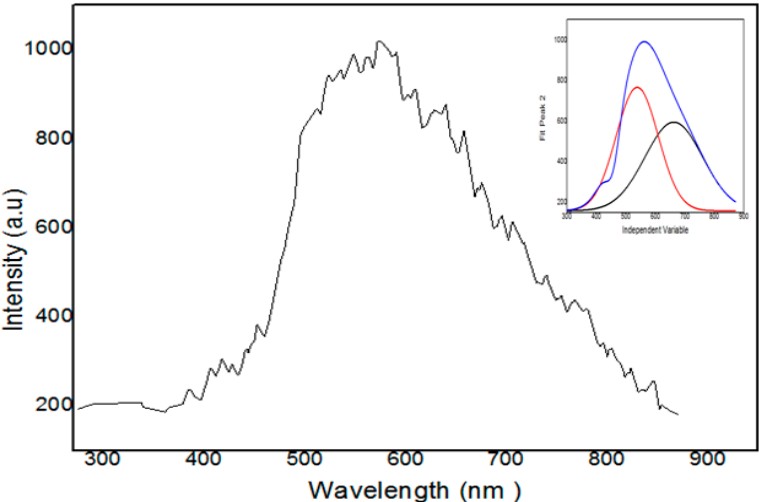

**Figure 11.** Electroluminescence of the cross-section of the $In_{0.1}Ga_{0.9}N$ LED.

Figure 12 shows the room temperature I–V curve (under dark) measured by two electrodes Potentionstat Auto Lab. 70,708 with the positive electrode of the electric source connected to the Au metal contact upon the $In_{0.1}Ga_{0.9}N$ LED. The results show that the diode has rectifying current characteristics with a current density of $1.2 \times 10^{-7}$ A/cm$^2$ for the diode area of 0.25 cm$^2$ [42].

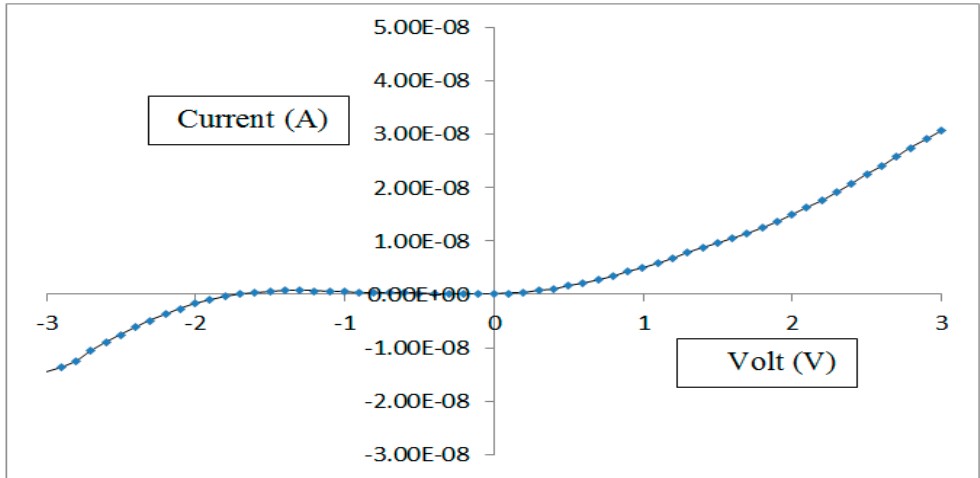

**Figure 12.** I–V curve of the cross-section of the $In_{0.1}Ga_{0.9}N$ LED.

## 4. Conclusions

This research focused on the growth of $In_xGa_{1-x}N$ as bulk alloys and thin films which work as an active layer in LED devices. The alloy growth was carried out in a programmable muffle using an evacuated silica tube with appropriate amounts of In and Ga metals under high purity ammonia gas. The optimized growth temperature of the bulk alloy was 950 °C with a strong PL peak energy of 2.65 eV at composition (x) = 0.1. The work also includes the doping of $In_{0.1}Ga_{0.9}N$ by the crystal growth technique for both n and p type using Zn and Mg, respectively. XRD was used to detect the growth of the grown bulk alloys and thin films exhibiting high crystallinity and no phase separation in most of them. Additionally, the novelty of this work is the use of a glass substrate, which is cheaper than sapphire. The LED device was glass/GaN/$In_{0.1}Ga_{0.9}N$:Zn/$In_{0.1}Ga_{0.9}N$/$In_{0.1}Ga_{0.9}N$: Mg, coated by Au at the p and n type layers which worked as metal electrodes. The $In_{0.1}Ga_{0.9}N$-based white LED layer structure was fabricated using a thermal vacuum evaporator. The output emission was a white light and was verified by EL spectroscopy. We believe that this study, which involved preparing $In_{0.1}Ga_{0.9}N$ as an active layer of blue emission and generating a white light with $In_{0.1}Ga_{0.9}N$ LEDs, offers a way to realize a low-cost technique for such highly needed devices globally.

**Author Contributions:** Conceptualization, A.E.-H.B.K. and M.F.; Investigation, A.E.-H.B.K., M.F. and S.G.; Methodology, M.F., S.G.; Supervision, A.E.-H.B.K., M.F. and Y.B.; Visualization, Y.B. and A.A.B.; Writing—review & editing, A.E.-H.B.K., M.F. and S.G.

**Funding:** This research received no external funding.

**Acknowledgments:** This work was carried out at City for Scientific Research and Technological Applications (SRTA-City) and collaboration research with the National Institute of Laser Enhanced Science.

**Conflicts of Interest:** The authors declare no conflict of interest.

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
