# Peer review of "Synthesis of Nanostructure InxGa1−xN Bulk Alloys and Thin Films for LED Devices"

_photonics, doi:10.3390/photonics6020044_

Reviewer 1 Report

 The author present a study about Synthesis of Nanostructure InxGa1-xN Bulk Alloys and Thin Films for LED Devices. The article is clear, well written.  The experimental part is very clearly described, making  future reproduction of this technique by other interested groups possible.

Even the optical characterization is performed exhaustively and scrupulously, with many details. I have no particular criticism to move to this article.

It just seems to me that in the references there are no many recent works, as in the last two years there have been many works about growth of InGaN nanostructures, especially for LED devices (for example I can remind of recent work by Tsatsulnikov's group at Ioffe of St. Petersburg, or from OSRAM scientific division, if I remember correctly).

So I just would invite the authors to check and update the references.          

Author Response

Thank you for your review of our paper

Point 1: So I just would invite the authors to check and update the references.     

Response 1:

The references are updated and highlighted in the reference section.

Reviewer 2 Report

Comments:

Review of Synthesis of Nanostructure InxGa1-xN Bulk Alloys and Thin Films for LED Devices.

This manuscript describes the fabrication method of a new form of a light-emitting diode (LED) device. The method combines nanostructure bulk alloys and thin films of indium gallium nitride (InxGa1-xN) as an active layer. Their experimental data suggests a new detection for fabrication of such devices from the InxGa1-xN LED devices with simple and low cost techniques. This reviewer believes the study carries a fair amount of merits. However, the manuscript has a number of serious issues that make the current version unsuitable for publication in Photonics.

1.      In this method, the vapor phase epitaxial (VPE) growth of GaN:Zn was intensively studied at 1980s and the emission peaks from 2.6 eV to 1.8 eV were reported already. The details have been provided in the 1980 Journal of Applied Physics paper [Ref 1]. Compared to those papers, the manuscript isn’t really anything new, which becomes even less significant. The lack of technical novelty puts this manuscript certainly below the bar for publication in the Journal of Photonics.

2.      For checking from the grain size (Fig. 3.2 and Table 3.1) and emission spot (Fig. 3.6), it is very small and cannot be used in the industry.

3.      There is no data to see the uniformity and reproducing capability of these grown LEDs on layers.

[1] B. Monemar, O. Lagerstedt, and H. P. Gislason, “Properties of Zn-doped VPE-grown GaN. I. Luminescence data in relation to doping conditions” J. Appl. Phys, 1980, 51(1), 625-639.

Author Response

Thank you very much for reviewing our manuscript 

Point 1: In this method, the vapour phase epitaxial (VPE) growth of GaN:Zn was intensively studied at 1980s and the emission peaks from 2.6 eV to 1.8 eV were reported already. The details have been provided in the 1980 Journal of Applied Physics paper [Ref 1]. Compared to those papers, the manuscript isn’t really anything new, which becomes even less significant. The lack of technical novelty puts this manuscript certainly below the bar for publication in the Journal of Photonics.

Response 1:

Thank you for your comment. The methodology used in the mentioned reference [1] seems to be different. They used Zn doping for GaN crystal growth. This was very nice work that concluded the proper applicability for such structure as acceptor dopant for the Led devices. The precursors of the materials in our work are Indium metal, Gallium metal and an ammonia gas which were placed in a vacuumed silica tube for temperatures between 750 ºC to 1150 ºC to prepare InxGa1-xN thin films. The Zn was used only for n type doped layer as explained in the experimental part. The main key in our method was how to bake these materials without needing more equipment. Only we needed a programmable muffle furnace under controlled temperature. The produced bulk alloys were then vacuum thermally deposited on glass substrates with some unique features such as high crystallinty, no separation phase, white light production and apparent good adhesion, which are highlighted in the manuscript in pages 3 and 4 (line.

Point 2: For checking from the grain size (Fig. 3.2 and Table 3.1) and emission spot (Fig. 3.6), it is very small and cannot be used in the industry.

Response 2:

 You are correct. Firstly we changed Table 3.1 to Table 3.3.2 and changed Fig (3.6) to Fig(3.8). We already mentioned particle size of values ranged from 35 nm at x=0.1 to 22 nm at x=0.5 instead of grain size in Table 3.3.2. These rates are coinciding with the value calculated from XRD according to Scherer's equation. The grain size averaged is 500 nm at x = 0.1, 0.3 and decreased to about 290 nm at x = 0.5, 0.7 and 0.9. This is highlighted in the manuscript in page 8 (lines 236-238).

Point 3 There is no data to see the uniformity and reproducing capability of these grown LEDs on layers.

Response 3:

This manuscript concentrated on developing both bulk alloys and thin films grown on glass substrates. Many attempts have been done to optimize the preparation conditions (In% content, temperature, doping with n and p types, etc.). The InxGa1-xN thin film of x = 0.1 yielded a unique morphology of columnar structure for such active layer as shown in Figure 3.7. This is highlighted in page 11 (line 322). Also, it appears from the same figure the uniformity of all other layers of n and p type and buffer layer that the range of its thickness is much closed. The reproducibility of the layers will be demonstrated for the future work of such devices.

Reviewer 3 Report

The paper presents experimental results of LED fabrications. The authors propose to use glass substrate and vary In-composition of InGaN to generating white LED light.

However, it is a little hard to evaluation the technique content of the paper. Because the paper’s English and structure are needed to be improved. Here are some examples, but not limit to the list below. The authors have to do a major revision of the paper and pay attention to all the details to make it  a reader friendly paper.

1.       Introduction :

Page 2 line46-49 is not a valid sentence. Please check all the paper for similar grammatical issues.

It is better to have two paragraphs in Introduction; it is not comfortable to read a giant paragraph with several meaning in it. At lease, the introduction can be separated into two parts around page 2 line 59.

2.       Fig 2.1 has two pictures. The first one has a (b)-label. I have no idea what it means. The second one does not have a label. In addition, the explanation in the paper is not clear as well.

3.       In Fig. 2.2, Temperature 750, 950, and 1150 should not be on the same level in the chart.

4.       There is no Fig. 2.3 in the paper, which is mentioned in page 3 line 87.

5.       Page 4 line 129 XRD should be spelled out as X-ray diffraction

6.       The Table 3.2 is in page 6 and Table 3.1 in page 7. Moreover, there is Table 3.2 in page 8 again. It is very confusing and hard to follow.

7.       Page 6 is section 3.3 and 3.3.1. However, Page 7 is section 3.2.2. Then on page 8 is section 3.3 and 3.3.1 again! How awkward.

8.       Page 8 line 262 Table 3.3 may be in page 9 where there is a table with no table title.

9.       Page 8 line 268 the calculation is not based on Table 3.1

10.   Page 9 figure 3.3 caption is overlap with the figure.

…..

One or two typo and grammatical issues is understandable. The paper in current format badly needs a major revision.

Author Response

Thank you very much for reviewing our manuscript

Point 1: Page 2 line46-49 is not a valid sentence. Please check all the paper for similar grammatical issues.

It is better to have two paragraphs in Introduction; it is not comfortable to read a giant paragraph with several meaning in it. At lease, the introduction can be separated into two parts around page 2 line 59.

Response 1: We checked the Grammars all over the manuscript and highlighted in red colour. The introduction was separated into two parts and rewrites the phase separation part.

Point 2: Fig 2.1 has two pictures. The first one has a (b)-label. I have no idea what it means. The second one does not have a label. In addition, the explanation in the paper is not clear as well.

Response 2

The first one on the left hand side has been changed from (b) to (a). The second picture is labelled by (b).The picture (a) is a schematic picture of the silica tube used in the experiment which filled with indium metal, gallium metal and ammonia gas which put in a furnace under different temperatures. The second picture (b) is a photograph of the final shape of the prepared material inside the silica tube which indicates to the growth of InxGa1-xN.This is highlighted in page 3 line 93.

Point 3: In Fig. 2.2 Temperature 750, 950, and 1150 should not be on the same level in the chart.

Response 3

Fig (2.2) changed to three levels as seen in the Figure 2.2 (page3) line 96.

Point 4: There is no Fig. 2.3 in the paper, which is mentioned in page 3 line 87.

Response 4

You are Right, sorry for that. Fig 2.3 is deleted from the text.

Point 5:

Page 4 line 129 XRD should be spelled out as X-ray diffraction

Response

Ok, we re-write it in page 4 line114

Point 6:

The Table 3.2 is in page 6 and Table 3.1 in page 7. Moreover, there is Table 3.2 in page 8 again. It is very confusing and hard to follow.

Response 6

It is modified from Table 3.2 to Table 3.2.1 in page 6 line172, Table 3.1to Table 3.3.1in page 7 line 205 and the repeated table number 3.2 to 3.3.2 is page 8  line 240 .

Point 7

Page 6 is section 3.3 and 3.3.1. However, Page 7 is section 3.2.2. Then on page 8 is section 3.3 and 3.3.1 again! How awkward.

Response 7

You are correct, Section 3.3 and 3.3.1 in page 6 line 137 and section 3.3.2in page 7 line 207. The repeated sections 3.3.and 3.3.1 were changed to 3.4 and 3.4.1 on page 8 line242 and line 243 respectivelly.

Point 8

    Page 8 line 262 Table 3.3 may be in page 9 where there is a table with no table title.

Response 8

 Ok it is changed from Table 3.3 to Table 3.4.1 line 269 pages 9,

Point 9

    Page 8 line 268 the calculation is not based on Table 3.1

Response 9

Ok it is checked and modified to Table [3.4.1] page 9 line 269

Point 10

 Page 9 Figure 3.3 caption is overlap with the figure.

Response 10

Ok it is changed from Figure 3.3 to Figure 3.5 in page 9 line 299

…..

One or two typo and grammatical issues is understandable. The paper in current format badly needs a major revision.

Response

The grammars are checked through the manuscript and highlighted in red

Round  2

Reviewer 2 Report

1. Showing an EL image of the In0.1Ga0.9N LED is insufficient to claim that its light emitting characterization has been performed. It could be improved by adding some figures of current voltage (I-V) or/and light-current (L-I) characteristics for this structure and text correspondent explaining in this section.

2. In your EL spectrum (Fig. 3.9), I see three decomposed sub Gaussian line shapes and their peaks (426 nm, 548 nm and 599 nm) from the spectrum. However, the crucial accompanying explanations of the 599nm peak is missing. Some explanation about the peak at the level of physics could be inserted in the main manuscript.

3. The efficacy of this LED is missing. Although the authors presented a detailed study, the reader is not convinced that the authors managed to obtain white light emission, especially by missing the luminous efficacy.

Author Response

Thank you very much for you revision

Response to Reviewer 2 Comments

Point 1: Showing an EL image of the In0.1Ga0.9N LED is insufficient to claim that its light emitting characterization has been performed. It could be improved by adding some figures of current voltage (I-V) or/and light-current (L-I) characteristics for this structure and text correspondent explaining in this section.

Response1

Thank you very much. It is added at page 13 lines[365-369] as I-V relationship for In0.1Ga0.9N  LED strcture

Figure (3.10): I-V curve of the cross section of In0.1Ga0.9N LED.

Figure (3.10) shows the room temperature I-V curve measured by two electrodes Potentionstat Auto Lab. 70708 with the positive electrode of the electric  source connected to the Au metal contact upon the In0.1Ga0.9N LED. The results shows that diode has a rectifying current characteristics with current density of 1.2x10-7 A/cm2 for the diode area of 0.25 cm2 [42].

Point 2. In your EL spectrum (Fig. 3.9), I see three decomposed sub Gaussian line shapes and their peaks (426 nm, 548 nm and 599 nm) from the spectrum. However, the crucial accompanying explanations of the 599 nm peak is missing. Some explanation about the peak at the level of physics could be inserted in the main manuscript.

Response 2

It checked at page 13 lines [355-369].

According to the InxGa1-xN structure, a blue emission originated from a recombination in the InxGa1-xN: Mg deposited on InxGa1-xN layer shifted toward a short wavelength (599 nm). Since that the mobility of electrons is faster than holes, therefore, electrons are injected from the n- InxGa1-xN: Zn side, through the InxGa1-xN layer, to the p- InxGa1-xN: Mg side and a little recombination occurred in the n- InxGa1-xN and InxGa1-xN layers. So the blue emission peak at 599 is observed [41].

Point 3:- The efficacy of this LED is missing. Although the authors presented a detailed study, the reader is not convinced that the authors managed to obtain white light emission, especially by missing the luminous efficacy.

Response 3

I think this point will be covered in details in the next publication for such very promising device.

Reviewer 3 Report

Page 2 line 46-50 is still needed to be revised.

Author Response

Thank you very much for your revision

Point 1

Page 2 line 46-50 is still needed to be revised

Response 1

It revised at lines [47-50] page2.

The difficulties in InxGa1-xN growth are mainly due to very high equilibrium vapor pressures (EVPs) of nitrogen over InN and a large lattice mismatch between InN and GaN. Additionally, the large lattice mismatch between InN and GaN resulted in highly strained InxGa1-xN alloys, which make phase separation is a major concern